# Utility of VIDAS^®^ Dengue Diagnostic Assays to Differentiate Primary and Secondary Dengue Infection: A Cross-Sectional Study in a Military Hospital from Colombia

**DOI:** 10.3390/tropicalmed10020040

**Published:** 2025-01-29

**Authors:** Andrés E. Prieto-Torres, Leidy J. Medina-Lozano, Juan David Ramírez-Ávila, Álvaro A. Faccini-Martínez

**Affiliations:** 1Internal Medicine Department, Hospital Militar Central, Bogotá D.C 110231, Colombia; andres08032018@outlook.com (A.E.P.-T.); leidymedinal88@gmail.com (L.J.M.-L.); 2Infectious Disease Department, School of Medicine, Universidad Nacional de Colombia, Bogotá D.C 111321, Colombia; 3Statistics Department, Universidad Nacional de Colombia, Bogotá D.C 111321, Colombia; juramireza@unal.edu.co; 4Infectious Disease Department, Hospital Militar Central, Bogotá D.C 110231, Colombia; 5School of Medicine, Universidad Militar Nueva Granada, Bogotá D.C 110231, Colombia

**Keywords:** dengue fever, antibodies, diagnostic test kit, antigen

## Abstract

This study aimed to assess the diagnostic utility of VIDAS^®^ DENGUE NS1 Ag and anti-DENV IgM and IgG assays in parallel for an early and accurate diagnosis and classification of dengue virus (DENV) infection. For this retrospective cross-sectional study, 190 patients with suspected dengue were tested using VIDAS^®^ NS1, IgM, and IgG assays, requested in parallel, regardless of symptom onset timing, and classified into primary and secondary infections. Results were analyzed to determine diagnostic accuracy and correlation with disease severity. Parallel testing effectively differentiated between primary and secondary DENV infection. Secondary dengue cases with warning signs showed significantly elevated IgG levels (*p* = 0.026). Notably, NS1-negative (possible secondary cases) had higher IgM and IgG levels than NS1-positive cases (*p* < 0.01), suggesting that NS1 negativity might indicate an amplified immune response. In conclusion, VIDAS^®^ Dengue diagnostic assays not only enhance the diagnostic accuracy of dengue infection but also offer valuable insights into serological patterns, especially in secondary cases. These assays could be used not only to confirm diagnosis but also to stratify patients by risk, particularly in cases of secondary dengue, where IgG levels might indicate a higher risk for severe outcomes.

## 1. Introduction

Dengue infection poses a significant global public health threat [1]. The dengue virus (DENV) belongs to the *Flaviviridae* family, which includes over 70 human pathogens, including yellow fever virus, Japanese encephalitis, and West Nile viruses [2]. Transmitted by *Aedes aegypti* and *Aedes albopictus* mosquitoes, dengue is an arboviral infection exhibiting a spectrum of diseases, from mild to severe [3]. Treatment focuses on supportive care, emphasizing timely diagnosis and monitoring of clinical progression [4].

Direct diagnosis of DENV infection relies on detecting circulating DENV RNA via reverse-transcription PCR (RT-PCR) and/or the viral non-structural protein 1 (NS1) antigen using immunoassays within the first 5–7 days of symptom onset [5]. Subsequent detection of anti-DENV immunoglobulin M (IgM) and G (IgG) via enzyme-linked immunosorbent assays (ELISA) or rapid diagnostic immunochromatographic tests (RDT) confirms primary or secondary DENV infection [6,7]. Combining these markers enhances diagnostic sensitivity, extending the time window for diagnosis of DENV infection [8].

NS1 antigen detection assays offer a simplified diagnostic approach during dengue’s acute phase by identifying viral antigens in whole blood. However, pre-existing IgG-viral immune complexes may compromise NS1 in secondary infections [9]. ELISA and RDT targeting NS1 protein detect high antigen levels up to nine days post-symptom onset in primary and secondary infections [10]. These tests are cost-effective, easy to perform, and effective for confirming infection, although sensitivity is lower than viral isolation or RNA detection methods [10].

Acquired immunity to DENV involves producing immunoglobulins (IgM, IgG, and IgA), targeting the envelope protein. The immune response intensity varies between primary and secondary infections [11,12,13]. Primary infections exhibit IgM detection around five days post-infection, followed by IgG detection after 10–15 days. Secondary infections show earlier IgM appearance at lower titers, while IgG titers surge rapidly [14]. Hemagglutination inhibition assays yield titers of up to 640 in primary infections and 1280 in secondary infections [15]. Primary infections elicit robust IgM responses and higher specificity than secondary infections. IgA- and IgE-based assays have also been previously used, though their diagnostic utility has not been validated [16].

The VIDAS^®^ Dengue Diagnostic Assays is a fully automated immunoassay for rapid diagnosis of DENV infections. This assay offers parallel (NS1-IgM-IgG) or independent (NS1, IgM, or IgG) operation, with sensitivity and specificity of 87.3–91.9% and 86.9–100%, respectively, providing clear positive/negative results within 40–60 min [8,17].

This retrospective cross-sectional study evaluated the diagnostic utility of the VIDAS^®^ Dengue Diagnostic Assays in differentiating between primary and secondary DENV infections among patients at a reference military hospital in Colombia while also exploring the temporal applicability of the serological test. Additionally, a statistical analysis was conducted to assess the performance of the assay as a potential prognostic marker for dengue with warning signs and to evaluate the plausibility of using NS1 as a marker of confirmed infection in patients with secondary dengue virus infection.

## 2. Materials and Methods

### 2.1. Patients and Samples

A total of 190 medical records were analyzed from patients with suspected DENV infection who presented between June 2023 and October 2024 at Hospital Militar Central, a reference military hospital in Bogotá, Colombia. All patients met the World Health Organization’s criteria for a probable dengue case and were evaluated by the Internal Medicine service in the emergency department. According to WHO classification, a probable dengue case is defined by a combination of ≥2 clinical findings in a febrile person who lives in or has traveled to (in the last 14 days) a dengue-endemic area. Clinical findings include nausea, vomiting, rash, aches and pains, a positive tourniquet test, leukopenia, or any warning signs.

### 2.2. Study Design and Definitions

This cross-sectional study aimed to assess the diagnostic utility of VIDAS^®^ Dengue Diagnostic Assays (bioMérieux SA, Marcy-l’Étoile, France) in distinguishing between primary and secondary DENV infections. The study focused on parallel detection of the DENV NS1 antigen (VIDAS^®^ DENGUE NS1 Ag), anti-DENV IgM (VIDAS^®^ Anti-DENGUE IgM) and IgG antibodies (VIDAS^®^ Anti-DENGUE IgG). The VIDAS^®^ Dengue Diagnostic Assays are automated two-step immunoassays. The VIDAS^®^ DENGUE NS1 Ag detects the NS1 antigen across all four DENV serotypes. The VIDAS^®^ Anti-DENGUE IgM and IgG detect IgM and IgG antibodies, respectively, with the ability to recognize antigens from all four DENV serotypes due to the inclusion of a recombinant tetravalent EDIIIT2 protein, which comprises the antigenic envelope domain III of each serotype [17,18]. Values of NS1-IgM-IgG ≥ 1 were considered positive, according to the manufacturer’s instructions.

For the initial analysis of the results, the criteria of positivity for primary infection were NS1 Ag positivity alone OR NS1 + IgM positivity OR IgM positivity + IgG/IgM ratio < 1.1 [19]. For secondary infections, the criteria of positivity were IgG positivity alone OR IgM positivity + IgG/IgM ratio ≥ 1.1 [19]; the diagnosis was confirmed if the NS1 Ag was positive, and it was probable/possible if the NS1 Ag was negative. This distinction was made with the sample at the entry of the patient, between days 1 to 10 after the beginning of symptoms.

In the second part of the study, for the analytical assessment of the proposed hypotheses, we compared the IgM, IgG, and IgG/IgM ratio serological results in the secondary dengue group between cases classified as confirmed (NS1 positive) and possible (high levels of IgG antibodies without NS1 antigen and low/negative level of IgM antibodies)

### 2.3. Statistical Analyses

As previously described, the first part of the study involves a description and analysis of the data obtained from the review of medical records of patients with suspected DENV infection (See Appendix A). In the second part, we compared the data in the “primary dengue infection without warning signs” category (42 patients) with those in the “primary dengue infection with warning signs” category (34 patients), focusing on IgM, IgG, and IgG/IgM ratio values (See Appendix A).

The Shapiro-Wilk test was used to determine whether the data followed a normal distribution. Since neither group exhibited a normal distribution, the Mann-Whitney U test (also known as the Mann-Whitney-Wilcoxon test or Wilcoxon rank-sum test) was applied to compare independent groups and identify significant differences in their distributions (See Appendix A).

This same process was used to compare the “secondary dengue infection without warning signs” category (31 patients) versus the “secondary dengue infection with warning signs” category (34 patients) and the categories “secondary dengue infection with positive NS1 antigen” (70 patients) versus “possible secondary dengue infection” (42 patients) (See Appendix A).

All hypothesis testing and calculations were performed using Python version 3.10.12, utilizing the *stats* library. A *p*-value of 0.05 or less was considered statistically significant (See Appendix A).

## 3. Results

### 3.1. Patients’ Characteristics

Out of the 190 reviewed suspected cases of dengue infection, 78 were classified as primary infections and 112 as secondary infections. Among the primary dengue cases, one presented a negative NS1 antigen, likely due to the timing of the test. However, IgM values and the IgG/IgM ratio confirmed its classification as a primary infection. Of the primary dengue cases, 58.4% were male and 41.6% female, with a mean age of 37.7 years. The average number of days from symptom onset to serological diagnosis was 4.93 days. The mean NS1 value was 80.67, with a median of 92. The mean and median values for IgM and IgG were 13.88 and 2, and 1.45 and 0, respectively, with a mean IgG/IgM ratio of 0.083, consistent with expected serological findings for a primary dengue infection.

For the secondary dengue infections, we classified them as possible or confirmed based on NS1 antigen results. Among the 70 confirmed secondary dengue cases, 62.86% were male, with a mean age of 40 years and an average of 5.05 days from symptom onset to diagnosis. The mean NS1 value was 54.5, while mean IgM and IgG values were 1.54 and 30.5, respectively, with an average IgG/IgM ratio of 18.89. Our findings align with the literature on secondary infections, which typically show an early significant IgG elevation with a minimal or attenuated IgM increase.

In the 42 possible secondary dengue cases, a higher proportion were male (78.57%), with a mean age of 31.7 years and an average of 6 days to diagnosis. The mean and median values for IgM and IgG were 5.78 and 2, 57.07 and 59.5, respectively, with an average IgG/IgM ratio of 18.89. Patient characteristics are summarized in Table 1.

### 3.2. Diagnostic Performance of VIDAS^®^ Assays in Relation to Time Elapsed from Symptom Onset to Diagnosis

Evaluating the results of simultaneous diagnostic requests for NS1, IgM, and IgG tests, based on the time elapsed from symptom onset to the point of testing, we performed serological diagnosis and classification using established standard timelines for test requests. We found that regardless of whether these tests are conducted in parallel on or before the fifth day of illness versus afterward, this approach allows for timely diagnosis of dengue virus infection. Moreover, it enables accurate classification of cases as primary or secondary infections, independent of timing, as shown in Figure 1.

### 3.3. Utility of VIDAS^®^ Assays as Predictors of Dengue with Warning Signs

When comparing the median results obtained from cases of primary and secondary dengue, classified as with or without warning signs (based on internationally defined criteria), no statistically significant differences were observed in the serology results for primary dengue cases. However, upon evaluating secondary dengue cases, serum IgG levels were significantly higher in patients classified with warning signs compared to those without warning signs (*p* = 0.026). See Table 2.

### 3.4. Utility of VIDAS^®^ Assays as Predictors of Confirmed Secondary Dengue Virus Infection

Finally, we assessed the hypothesis regarding the performance of IgM, IgG serological assays, and the IgG/IgM ratio in predicting confirmed secondary dengue infection. We analyzed the groups of possible secondary dengue infection (NS1−) and confirmed secondary dengue infection (NS1+), as shown in Table 1. Our analysis revealed that both IgM (*p* = 0.0011) and IgG levels (*p* = 0.0000058) were significantly elevated in cases where NS1 test results were negative. This finding supports our suspicion that NS1 negativity in secondary dengue infection correlates with a heightened immune response. Thus, NS1 status alone is not a reliable parameter for predicting confirmed dengue infection in the absence of RT-PCR. These results are summarized in Table 3.

## 4. Discussion

This study evaluated the diagnostic utility of VIDAS^®^ Dengue diagnostic assays—DENV NS1 antigen (VIDAS^®^ DENGUE NS1 Ag), anti-DENV IgM (VIDAS^®^ Anti-DENGUE IgM), and anti-DENV IgG (VIDAS^®^ Anti-DENGUE IgG)—when performed in parallel, regardless of the time elapsed since symptom onset, for diagnosing and classifying dengue as either primary or secondary.

The DENV contains an 11 kb single-stranded RNA genome encoding a polypeptide that gives rise to seven non-structural and three structural proteins [19]. Among these, NS1 is a 48 kDa glycoprotein that appears in several forms—membrane-associated (NS1m), vesicle-associated within cells, or as a secreted extracellular molecule (NS1s) [20]. NS1’s extracellular presence correlates with peak viremia and disease severity, especially in secondary infections, due to its role in immune activation and endothelial disruption [21]. NS1 facilitates viral replication, activates Toll-like receptor 4 (TLR-4) in monocytes and macrophages, and triggers pro-inflammatory cytokines, which contribute to the vascular permeability linked with severe dengue [22,23].

Most circulating antibodies after infection target various epitopes on the E and prM proteins but do not neutralize the virus [24]. Instead, they mediate “antibody-dependent enhancement” (ADE), allowing viral entry into immune cells via Fc gamma receptors [25]. This unique mechanism increases the risk of severe disease in secondary heterotypic infections, where the time interval between infections appears to influence complication risk [26].

Our findings align with the natural course of dengue illness, where, for primary cases, IgM appears early—between days 3 and 5 after fever onset in around 50% of hospitalized patients, with a sensitivity of 90% and specificity of 98% when taken after day five [9,13,14,15,27]. ELISA-based serological assays commonly target the E protein of all four dengue serotypes, aiming for broad detection [9,13,14,15,27]. VIDAS^®^ assays offer critical insights into the optimal diagnostic timing through NS1, IgM, and IgG, capturing variations at different infection stages. Rapid immunochromatographic tests that detect NS1/IgM/IgG were simultaneously reported to have a general sensitivity of 91% and specificity of 96% [27,28,29]. According to our results, VIDAS^®^ assays have been shown to effectively support timely dengue diagnosis when conducted in parallel, irrespective of the testing day as well as, and importantly, VIDAS^®^ assays allowed accurate classification of primary and secondary dengue cases beyond the standard early diagnostic window.

With the above, our results contrast with the algorithm for laboratory confirmation of dengue cases proposed by the Pan American Health Organization (PAHO), which suggests performing NS1 antigen detection only during the first 5 days after the onset of symptoms, detection of IgM ≥ 6 days from the onset of symptoms, and a limited diagnostic value of IgG measurements [30]. Thus, the VIDAS^®^ Dengue Diagnostic Assays, requested in parallel, would be an alternative to the PAHO algorithm for confirmation of dengue cases and its classification as primary or secondary, regardless of symptom onset timing.

For primary dengue cases, no significant differences were found in serological results between patients with and without warning signs. However, secondary dengue cases displayed significantly higher IgG levels in patients with warning signs, suggesting a potential role for IgG values in identifying individuals at risk of severe dengue. Additionally, our data revealed that NS1-negative secondary cases had significantly elevated IgM and IgG levels compared to NS1-positive cases, supporting the hypothesis that NS1 negativity might signal a robust immune response in secondary infections, which are often associated with more severe disease [31]. This finding suggests that NS1 status alone may be insufficient to confirm secondary infections in the absence of RT-PCR and that elevated IgM and IgG levels may serve as more reliable indicators of confirmed secondary dengue infection in these patients. Nevertheless, as a clear limitation of our study, in the absence of confirmatory markers (PCR, DENV serological tests other than VIDAS^®^, serological tests against other arboviruses), our retrospective design does not allow to conclude that the 42 patients with NS1 negative antigen were dengue secondary cases. Moreover, regarding the NS1 antigen, it exhibited a satisfactory positive predictive value of dengue, but its negative predictive value cannot be estimated in the absence of a true biological marker for distinguishing an NS1-negative secondary dengue infection from a past infection in a patient with a clinical dengue-like syndrome.

Although our findings are derived from a retrospective analysis in a cross-sectional study, they are innovative and suggest hypotheses to be tested in future studies with larger sample sizes and different methodological designs. Such studies could confirm our conclusions, especially regarding the establishment of specific IgM and IgG cut-off points for confirmed secondary dengue infections, ideally using simultaneous RT-PCR as a reference standard.

## 5. Conclusions

VIDAS^®^ Dengue diagnostic assays not only enhance the diagnostic accuracy of dengue infection but also offer valuable insights into serological patterns, especially in secondary cases. These assays could be used not only to confirm diagnosis but also to stratify patients by risk, particularly in cases of secondary dengue, where IgG levels might indicate a higher risk for severe outcomes.

## Figures and Tables

**Figure 1 tropicalmed-10-00040-f001:**
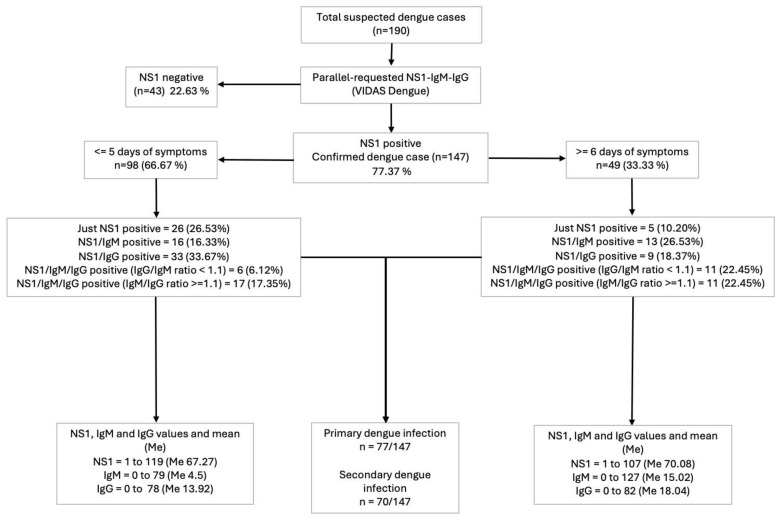
Classification of cases as primary or secondary dengue infections.

**Table 1 tropicalmed-10-00040-t001:** Patients’ characteristics.

Category	Patient’s Number	Men N (%)	Women N (%)	Statistical Parameter	Day to Diagnosis	Age	NS1 *	IgM *	IgG *	IgG/IgM Ratio
Confirmed primary dengue infection	77	45 (58.44)	32 (41.56)	M	4.93	37.74	80.67	13.88	1.45	0.083
				SD	2.01	18.7	24.85	22.81	5.12	0.197
				Min	1	18	9	0	0	0
				Me	5	32	92	2	0	0
				Max	9	92	119	127	39	1
Probable primary dengue infection	1	1 (100)	0 (0)	M	6	32	0	131	73	0.557
				SD	-	-	-	-	-	-
				Min	-	-	-	-	-	-
				Me	-	-	-	-	-	-
				Max	-	-	-	-	-	-
Confirmed secondary dengue infection	70	44 (62.86)	26 (37.14)	M	5.05	40.08	54.5	1.54	30.5	17.65
				SD	1.72	17.7	38.3	2.71	29.01	16.9
				Min	1	18	1	0	1	1.22
				Me	5	39	57	0	15.5	12
				Max	10	84	118	13	82	75
Possible secondary dengue infection	42	33 (78.57)	9 (21.43)	M	6.11	31.71	0	5.78	57.07	18.89
				SD	1.5	11.43	-	10.31	20.5	14.6
				Min	3	17	-	0	6	1.4
				Me	6	32.5	-	2	59.5	17.33
				Max	9	75	-	49	82	54

N: number; M: average; SD: standard deviation; Min: minimum value; Me: median; Max: Maximum value; * All values are reported in the standard Index Value.

**Table 2 tropicalmed-10-00040-t002:** Dengue VIDAS^®^ Assays as Predictors of Dengue with Warning Signs.

Category	IgM *p*-Value	IgG *p*-Value	IgG/IgM Ratio *p*-Value
Primary dengue infection without warning signs	0.18	0.66	0.80
Primary dengue infection with warning signs			
Secondary dengue infection without warning signs	0.6	0.026	0.0690
Secondary dengue infection with warning signs			

**Table 3 tropicalmed-10-00040-t003:** Dengue VIDAS^®^ Assays as Predictors of Secondary Dengue.

Category	IgM *p*-Value	IgG *p*-value	IgG/IgM Ratio *p*-Value
Possible secondary dengue infection	0.00112	0.0000058	0.560
Confirmed secondary dengue infection			

## Data Availability

Clinical data is unavailable due to privacy restrictions.

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
