# Peer review of "Utility of VIDAS® Dengue Diagnostic Assays to Differentiate Primary and Secondary Dengue Infection: A Cross-Sectional Study in a Military Hospital from Colombia"

_tropicalmed, 2025, doi:10.3390/tropicalmed10020040_

Round 1

Reviewer 1 Report

Comments and Suggestions for Authors

An interesting analysis of primary, secondary, and probably Dengue using NS1, IgM, and IgG levels by testing for all three simultaneously.

Comment:

1. As Dengue is often suspected based on clinical presentation, some discussion on clinical findings will help improve the impact of the paper.

2. What were the associated features of secondary dengue and its presentation, how long after the primary episode did it occur?

3. What was the association of either IgM or IgG with complications, if any?

Author Response

  1. As Dengue is often suspected based on clinical presentation, some discussion on clinical findings will help improve the impact of the paper.

 RESPONSE: We appreciate the comment, however the main of our manuscript was focused on the utility of VIDAS® Dengue Diagnostic Assays for differentiate primary and secondary dengue infection. Nevertheless, another manuscript will be prepared considering the same patients, in order to present the demographic, clinical and laboratory features of dengue in our military hospital.

  1. What were the associated features of secondary dengue and its presentation, how long after the primary episode did it occur?

 RESPONSE: We appreciate the comment, however the main of our manuscript was focused on the utility of VIDAS® Dengue Diagnostic Assays for differentiate primary and secondary dengue infection. Nevertheless, another manuscript will be prepared considering the same patients, in order to present the demographic, clinical and laboratory features of dengue in our military hospital.

 What was the association of either IgM or IgG with complications, if any?

RESPONSE: In our study we do not assess dengue complications. Nevertheless, our results shows that upon evaluating secondary dengue cases, serum IgG levels were significantly higher in patients classified with warning signs compared to those without warning signs (p = 0.026). Please see “3.3” section, and Table 2. 

Reviewer 2 Report

Comments and Suggestions for Authors

The manuscript by Prieto-Torres et al. provides an interesting overview of the use of the VIDAS Dengue Diagnostic Assays for the diagnosis of dengue in patients infected by this agent. Despite the fact that the study was retrospective and that no other molecular or serological technique was used to confirm the results, it stresses the potential ability of this kit to discriminate primary from secondary infections.

The main critics is the absence of alternative techniques to confirm the good results of the VIDAS technology. Indeed, the test was used for discriminating primary from secondary dengue infections and, in the same time, the authors conclude that these assays “enhance the diagnostic accuracy of dengue infection … offer valuable insights into serological patterns, especially in secondary cases.” In the absence of reference tests, it is not possible to be so affirmative. For instance, what is the specificity of a positive IgM test in the absence of NS1 positivity with the VIDAS technology? The only reference in this study is the clinical context that is only evoked in the Methods section by the sentence: “All patients met the World Health Organization’s criteria for a probable dengue case”.

This confusion between the use of the VIDAS tests for both classifying the patients and evaluating their performances must be corrected through the whole manuscript. Notably, the criteria of classification of patients must be defined much more clearly in the “Methods” section. For instance:

-          For primary infection, the criteria of positivity were: NS1 Ag positivity alone OR NS1+IgM positivity OR IgM positivity + IgG/IgM ratio < 1.1.

-          For secondary infections, the criteria of positivity were IgG positivity alone OR IgG/IgM ratio >= 1; the diagnosis was confirmed if the NS1 Ag was positive and it was probable if the NS1 Ag was negative.

The authors must explain more clearly why it is important to discuss the results according to the time of diagnosis of symptoms, as shown in Figure 1. Regarding this Figure, I am not sure it is a good idea to represent also primary and secondary cases on the same flowchart; otherwise, the same must be done for the 43 cases tested negative for NS1 Ag (1 primary and 42 secondary).

The results of paragraph 3.4 are also puzzling. It is said that a high level of anti-DENV antibodies is predictive of a “confirmed” secondary dengue. I do not think that this conclusion is valid. What is shown is that patients with a secondary dengue exhibit a higher level of antibodies (IgM and IgG) when the NS1 Ag is tested negative. But this finding is not surprising since the patients with NS1 Ag are able to make immune complexes probably at the origin of the decrease of the specific antibody levels. In the absence of kinetics to study the evolution of the different markers between these two groups of patients (maybe those with negative NS1 Ag were tested later than those with positive NS1 Ag), it is not possible to conclude on the signification of this difference, except for the evident explanation given just above. I suggest to remove this part of the study.   

A few minor points need also to be clarified (L for line):

-          L33: “Flaviviridae” in italics.

-          L35: “occasionally” for South America but “mainly” in other parts of the world as Europe.

-          L57-58: it is not useful to define values for antibody titers [which must not be expressed as ratios but as numbers (640 and 1280)].

-          L77-78: please recall here the clinical criteria used for including the patients as “probable” dengue.

-          L93-96: please explain clearly that this distinction was made at entry of the patient, between days 1 to 10 after the beginning of symptoms, according to Table 1.

-          Table 1: Use points instead of commas for numeral values.

-          Tables 2: Recall what was the statistical test that was used for these calculations

-          L183-197: Introduction rather than discussion.

-          L230: “could be useD”.

-          Supplementary materials. Is this appendix useful? The statistical paragraph explains clearly what was done. The details of calculations are not useful for the understanding of the results.  

As a whole, the manuscript needs major revision. The limitations of the study must be stressed much more clearly and the conclusions must be limited to the results shown in the paper.

Author Response

The main critics is the absence of alternative techniques to confirm the good results of the VIDAS technology. Indeed, the test was used for discriminating primary from secondary dengue infections and, in the same time, the authors conclude that these assays “enhance the diagnostic accuracy of dengue infection … offer valuable insights into serological patterns, especially in secondary cases.” In the absence of reference tests, it is not possible to be so affirmative. For instance, what is the specificity of a positive IgM test in the absence of NS1 positivity with the VIDAS technology? The only reference in this study is the clinical context that is only evoked in the Methods section by the sentence: “All patients met the World Health Organization’s criteria for a probable dengue case”.

 RESPONSE: We appreciate the comment. However, although an alternative diagnostic technique was not used, VIDAS® Dengue Diagnostic Assays has been validated in two previous studies, one of them a multicentre international study, with excellent sensitivity and specificity data for the diagnosis of dengue (Please see:  10.3390/diagnostics11071228; 10.3390/diagnostics13061137). Respectfully, we consider that these previous publications offer a reference that supports the adequate performance of VIDAS® Dengue Diagnostic Assays. In fact, we used these references at the "Introduction" section (references "8" and "17"). Thus, the objective of our studies was to demonstrate that, performing the three assays (NS1, IgM and IgG) in parallel, in addition to reaffirming the fact of confirming cases through the NS1 test, it was possible to classify and differentiate primary and secondary infections considering the results of IgM and IgG and their values. On the other hand, in order to offer more information regarding the sensitivity and specificity of VIDAS® Dengue Diagnostic Assays, these data are added to the text (Highlighted in lines 64-65). Finally, the phrase regarding the definition of a probable case of dengue, according to the WHO, was used to indicate the initial point of evaluation of patients, who underwent the three VIDAS assays in parallel. We expanded and specified the definition of a probable case of dengue (Highlighted in lines 77-82).

This confusion between the use of the VIDAS tests for both classifying the patients and evaluating their performances must be corrected through the whole manuscript. Notably, the criteria of classification of patients must be defined much more clearly in the “Methods” section. For instance:

- For primary infection, the criteria of positivity were: NS1 Ag positivity alone OR NS1+IgM positivity OR IgM positivity + IgG/IgM ratio < 1.1.

-  For secondary infections, the criteria of positivity were IgG positivity alone OR IgG/IgM ratio >= 1; the diagnosis was confirmed if the NS1 Ag was positive and it was probable if the NS1 Ag was negative.

RESPONSE: Thank you for the suggestions. In the “Methods” section, we defined much more clearly the criteria of classification of patients for primary and secondary infection, as review suggested (Highlighted in lines 96-100).

The authors must explain more clearly why it is important to discuss the results according to the time of diagnosis of symptoms, as shown in Figure 1. Regarding this Figure, I am not sure it is a good idea to represent also primary and secondary cases on the same flowchart; otherwise, the same must be done for the 43 cases tested negative for NS1 Ag (1 primary and 42 secondary).

RESPONSE: As suggested by the reviewer, we explain more clearly the importance of our results according to the time of diagnosis of symptoms (Highlighted in lines 214-220). On the other hand, the objective of presenting both primary and secondary cases in "Figure 1" is to show that regardless of the time at which the samples were taken (before or after day 5), it was possible to confirm cases and classify them into primary and secondary dengue. In addition, we do not consider presenting in the "Figure 1" the additional values ​​of the negative NS1 cases, since, at this point in the article, we only want to highlight the usefulness of the request in parallel to confirm cases and classify them into primary and secondary. The results of the negative NS1 were only used for subsequent analyses.

The results of paragraph 3.4 are also puzzling. It is said that a high level of anti-DENV antibodies is predictive of a “confirmed” secondary dengue. I do not think that this conclusion is valid. What is shown is that patients with a secondary dengue exhibit a higher level of antibodies (IgM and IgG) when the NS1 Ag is tested negative. But this finding is not surprising since the patients with NS1 Ag are able to make immune complexes probably at the origin of the decrease of the specific antibody levels. In the absence of kinetics to study the evolution of the different markers between these two groups of patients (maybe those with negative NS1 Ag were tested later than those with positive NS1 Ag), it is not possible to conclude on the signification of this difference, except for the evident explanation given just above. I suggest to remove this part of the study.

RESPONSE: As noted in the "Introduction" section, lines 45-49, secondary dengue infections often elicit a much more robust Ag-Ab response, which partially explains the negative NS1 antigen result. Therefore, while this finding may not be surprising, as pointed out by the reviewer, it is highly significant and has not been previously documented in prior studies. For this reason, we believe it is important to retain the result. The finding gains further relevance, as mentioned in the Discussion, within the Latin American context, where RT-PCR for dengue diagnosis is rarely available. Additionally, the kinetics of antibodies and antigens are not commonly understood by the majority of medical professionals who frequently face the challenge of diagnosing the disease accurately and early.

A few minor points need also to be clarified (L for line):

- L33: “Flaviviridae” in italics. 

RESPONSE: We corrected it, as suggested (Highlighted in line 33).

- L35: “occasionally” for South America but “mainly” in other parts of the world as Europe.

RESPONSE: We deleted "primarily" and "occasionally" (Highlighted in lines 34-35).

- L57-58: it is not useful to define values for antibody titers [which must not be expressed as ratios but as numbers (640 and 1280)].

RESPONSE: We corrected it, as suggested (Highlighted in line 57).

- L77-78: please recall here the clinical criteria used for including the patients as “probable” dengue.

RESPONSE: We corrected it, as suggested (Highlighted in lines 76-82).

- L93-96: please explain clearly that this distinction was made at entry of the patient, between days 1 to 10 after the beginning of symptoms, according to Table 1.

 RESPONSE: We corrected it, as suggested (Highlighted in lines 100-101).

-  Table 1: Use points instead of commas for numeral values.

RESPONSE: We corrected it, as suggested.

-  Tables 2: Recall what was the statistical test that was used for these calculations

RESPONSE: Refer to the Materials and Methods section and the supplementary materials, where the statistical method used is thoroughly explained and exemplified, as presented in the graphs.

- L183-197: Introduction rather than discussion.

RESPONSE: Respectfully, we considered these paragraphs as necessary to give meaning to the discussion we have later.

- L230: “could be useD”.

RESPONSE: We corrected it, as suggested Highlighted in line 241).

- Supplementary materials. Is this appendix useful? The statistical paragraph explains clearly what was done. The details of calculations are not useful for the understanding of the results. 

RESPONSE: Respectfully, we consider the supplementary material necessary in case readers require to verify the details of calculations.

As a whole, the manuscript needs major revision. The limitations of the study must be stressed much more clearly and the conclusions must be limited to the results shown in the paper.

RESPONSE: We corrected the manuscript, following the reviewer suggestions.

Round 2

Reviewer 2 Report

Comments and Suggestions for Authors

I regret that most of my remarks were not taken into consideration and that only formal changes were done by the authors. 

As for flu, dengue is not a clinical but a virological diagnosis. Dengue-like syndromes are frequent in tropical areas (the authors know that probably better than myself) and all dengue-like diseases are not dengue, especially in the context of Latin America where different arboviruses circulate at the same time in the same place (co-infections are frequent). I mention that because I disagree with the authors when they classify cases with high levels of IgG antibodies without NS1 antigen and low/negative level of IgM antibodies as "probable" secondary dengue. I agree with "possible" but not "probable". In an area where DENV is circulating, a dengue-like infection with a positive serology but without direct markers (NS1 antigen or positive RT-PCR) of dengue infection cannot be ruled out as "probable" dengue; it can also correspond to an infection with another pathogen associated to a past dengue infection. This could explain why patients with warning signs exhibited higher levels of IgG anti-DENV antibodies since most of these cases were "probably" true dengue infections, which could not be the case for patients with milder symptoms.

In the absence of confirmatory markers (PCR, DENV serological tests other than VIDAS, serological tests against other arboviruses)..., the retrospective design of the study does not allow to conclude that the 41 patients with NS1 negative antigen were "probable" dengue secondary cases. Regarding the NS1 antigen, it exhibited a satisfactory positive predictive value of dengue but its negative predictive value cannot be estimated in the absence of true biological marker for distinguishing an NSI-negative secondary dengue infection from a past infection in a patient with a clinical dengue-like syndrome. 

In conclusion, I continue to require change in the Results section with this clear limitation that no marker was available for evaluating the exact etiological diagnosis of the 42 patients with symptoms of dengue without IgM or NS1 positivity.

Author Response

I regret that most of my remarks were not taken into consideration and that only formal changes were done by the authors.

As for flu, dengue is not a clinical but a virological diagnosis. Dengue-like syndromes are frequent in tropical areas (the authors know that probably better than myself) and all dengue-like diseases are not dengue, especially in the context of Latin America where different arboviruses circulate at the same time in the same place (co-infections are frequent). I mention that because I disagree with the authors when they classify cases with high levels of IgG antibodies without NS1 antigen and low/negative level of IgM antibodies as "probable" secondary dengue. I agree with "possible" but not "probable". In an area where DENV is circulating, a dengue-like infection with a positive serology but without direct markers (NS1 antigen or positive RT-PCR) of dengue infection cannot be ruled out as "probable" dengue; it can also correspond to an infection with another pathogen associated to a past dengue infection. This could explain why patients with warning signs exhibited higher levels of IgG anti-DENV antibodies since most of these cases were "probably" true dengue infections, which could not be the case for patients with milder symptoms.

In the absence of confirmatory markers (PCR, DENV serological tests other than VIDAS, serological tests against other arboviruses)..., the retrospective design of the study does not allow to conclude that the 41 patients with NS1 negative antigen were "probable" dengue secondary cases. Regarding the NS1 antigen, it exhibited a satisfactory positive predictive value of dengue but its negative predictive value cannot be estimated in the absence of true biological marker for distinguishing an NSI-negative secondary dengue infection from a past infection in a patient with a clinical dengue-like syndrome.

In conclusion, I continue to require change in the Results section with this clear limitation that no marker was available for evaluating the exact etiological diagnosis of the 42 patients with symptoms of dengue without IgM or NS1 positivity.

 RESPONSE: Thank you for your suggestions. In the new revised version of the manuscript and in the supplementary materials, we change all "cases with high levels of IgG antibodies without NS1 antigen and low/negative level of IgM antibodies" as "possible" secondary dengue. In addition, in the "discussion" section we wrote a clear limitation of our study, as following: "Nevertheless, as a clear limitation of our study, in the absence of confirmatory markers (PCR, DENV serological tests other than VIDAS, serological tests against other arbo-viruses), our retrospective design does not allow to conclude that the 42 patients with NS1 negative antigen were dengue secondary cases. Moreover, regarding the NS1 antigen, it exhibited a satisfactory positive predictive value of dengue but its negative predictive value cannot be estimated in the absence of true biological marker for distinguishing an NS1-negative secondary dengue infection from a past infection in a patient with a clinical dengue-like syndrome."